# Hierarchical Image Tokenization for Multi-Scale Image Super Resolution

Isma Hadji [* 1]   Enrique Sanchez [* 1]   Adrian Bulat [1 2]   Brais Martinez [1]   Georgios Tzimiropoulos [1 3]

## Abstract

We introduce a multi-scale Image Super Resolution (ISR) method building on recent advances in Visual Auto-Regressive (VAR) modeling. VAR models break image tokenization into additive, gradually increasing scales, using Residual Quantization (RQ), an approach that aligns perfectly with our target ISR task. Previous works taking advantage of this synergy suffer from two main shortcomings. First, due to the limitations in RQ, they only generate images at a predefined fixed scale, failing to map intermediate outputs to the corresponding image scales. They also rely on large backbones or a large corpus of annotated data to achieve better performance. To address both shortcomings, we introduce two novel components to the VAR training for ISR, aiming at increasing its flexibility and reducing its complexity. In particular, we introduce a) a **Hierarchical Image Tokenization (HIT)** approach that progressively represents images at different scales while enforcing token overlap across scales, and b) a **Direct Preference Optimization (DPO) regularization term** that, relying solely on the (LR,HR) pair, encourages the transformer to produce the latter over the former. Our proposed HIT acts as a strong inductive bias for the VAR training, resulting in a small model (300M params vs 1B params of VARSR), that achieves state-of-the-art results without external training data, and that delivers multi-scale outputs with a single forward pass.

## 1. Introduction

Image Super-Resolution (ISR) is the task of generating a high-resolution image (HR) from a low-resolution (LR) version. Strong ISR methods typically rely on generative priors,

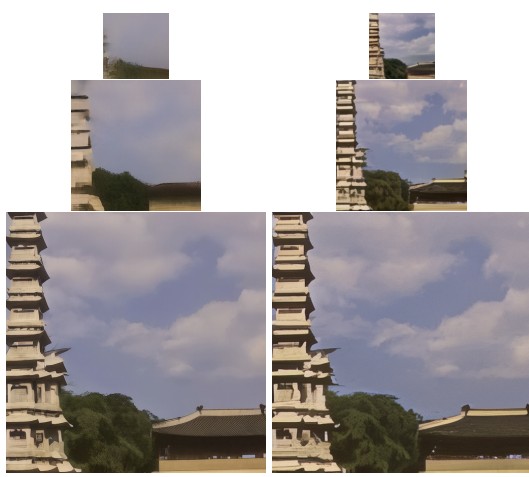

Figure 1: We propose a VAR-based Multi-scale SR method.**(Left)** Existing methods (Qu et al., 2025) can only target a single scale (e.g. $\times 4$). **(Right)** Instead we take full advantage of the next-scale prediction paradigm by introducing *Hierarchical Image Tokenization*. Using this approach our model can progressively upscale an image while keeping semantic consistency. Top-to-bottom images correspond to output of the models at scales $128 \times 128$, $256 \times 256$, $512 \times 512$, respectively, corresponding to scale factors of $\times 1$, $\times 2$ and $\times 4$.

either GAN-based (Ledig et al., 2017; Wang et al., 2021) or diffusion-based (Wang et al., 2024; Noroozi et al., 2024; 2025), to enforce that the output image falls within the manifold of natural images. Recently, Visual Auto-Regressive (VAR) models have shown competitive performance for image generation, providing an alternative source of generative priors (Sun et al., 2024; Tian et al., 2024; Han et al., 2025; Tang et al., 2025; Ma et al., 2024). We focus on the next-scale prediction variant due to its state-of-the-art image generation performance and the exceptional alignment between VAR pre-training and the ISR downstream task.

Previous works adopting the VAR formulation for the ISR task (Qu et al., 2025; Wei et al., 2025), do not fully exploit the next-scale prediction paradigm, as they are still limited to only upscale images to a predefined, fixed, scale factor. For example, VARSR (Qu et al., 2025) uses the same Residual Vector Quantization (RQVAE (Gray, 1984; Van Den Oord et al., 2017; Lee et al., 2022)) proposed in the

[*]Joint first authorship (alphabetical order) [1]Samsung AI Center Cambridge, UK [2]Technical University of Iasi, Romania [3]Queen Mary University of London, UK. Correspondence to: Enrique Sanchez <e.lozano@samsung.com>.

*Proceedings of the $43^{rd}$ International Conference on Machine Learning*, Seoul, South Korea. PMLR 306, 2026. Copyright 2026 by the author(s).

original VAR (Tian et al., 2024), which quantizes an image according to increasingly higher resolution levels. However, this RQVAE does not guarantee that intermediate scales can be decoded into valid images. This effect limits the derived ISR models to operate on fixed upsampling factors, while failing at intermediate scales as seen on the left side of Fig. 1. Instead, we introduce a **Hierarchical Image Tokenization (HIT)** approach in which the tokenization is applied sequentially to downsampled versions of the input image, while encouraging higher scales to reuse the tokens from lower scales. We show that by simply finetuning the vocabulary and decoder of an RQVAE trained for 512 resolution using our hierarchical representation, the tokenization produces a residual that can be mapped to intermediate image resolutions. In turn, the resulting VAR can target multi-scale ISR with a single forward pass, as shown in Fig. 1 on the right.

In addition to the lack of multi-scale formulation, previous methods either combine VAR models with large VLM models (Wei et al., 2025), or rely on large-scale carefully annotated data (Qu et al., 2025), to guide the ISR process and yield competitive results. In contrast, our formulation that leverages intermediate scales, allows us to yield competitive results, while using a relatively small model and standard training data. To further enhance quality, without relying on external guidance, we propose to train the VAR using **a regularization term** that is **based on Direct Preference Optimization** (Rafailov et al., 2023), which drives the model to "prefer" the sequence of HR tokens over those of the LR. Besides super-resolving flexibly at different scale factors, our simple 310M model, surpasses the much heavier counterparts, while relying on standard training data. Our main **contributions** can be summarized as follows:

- We propose the first *multi-scale* VAR-based approach for ISR by introducing a **Hierarchical Image Tokenization** (HIT), that can decode intermediate scales to the image space, keeping the semantic information across scales consistent. Interestingly, we observe that *our proposed HIT method acts as a strong inductive bias* for the VAR training, thus removing the need of large backbones to attain state of the art results.

- We introduce a simple **DPO-based regularization term** that, contrary to previous work, does not require collecting large-scale data with negative samples, or using additional VLM, simplifying the training.

## 2. Related Work

**Image Super Resolution**   Early methods for ISR mainly focused on developing efficient CNN-based architectures, using standard reconstruction and perceptual losses (Liu et al., 2023; Chen et al., 2024; Dong et al., 2014; Wang et al., 2015; Kim et al., 2016; Lim et al., 2017; Zhang et al.,

2018b). To improve image quality, GAN-based approaches were introduced to synthesize realistic features beyond interpolation and denoising (Wang et al., 2021; Ledig et al., 2017; Gu et al., 2020). GAN-based approaches are however hard to train and prone to produce hallucinations. The development of strong denoising diffusion models, with strong priors for text-to-image (T2I) and image-to-image (I2I) tasks greatly influenced the advancement in super resolution models (Wang et al., 2024; Noroozi et al., 2024; 2025; Mei et al., 2024; Yu et al., 2024; Wu et al., 2024).

**Autoregressive Image Generation**   Visual Autoregressive (VAR) models have very recently challenged the dominance of diffusion models for image generation, e.g. (Tian et al., 2024; Li et al., 2024; Voronov et al., 2025; Fan et al., 2025). Among these, next-scale auto-regressive image models (Tian et al., 2024; Tang et al., 2025; Voronov et al., 2025; Li et al., 2024; Fan et al., 2025) are particularly well suited for ISR. They typically rely on a variant of image tokenization (Van Den Oord et al., 2017; Esser et al., 2021) based on residual quantization (Lee et al., 2022), and are trained to iteratively regress the residuals across a sequence of monotonically increasing resolutions while conditioning on all previous scales. This scheme offers ideal alignment between pre-training and downstream (ISR) settings. The success of AR models for vision tasks has lately been extended to Image Restoration (IR, (Jiang et al., 2025; 2024; Zhu et al., 2025b)), a task with certain similarities to that of ISR. Varformer (Wang & Zhao, 2025) and RestoreVAR (Rajagopalan et al., 2026) exemplify the success of AR models in such downstream vision tasks.

**AR-based super resolution**   To our knowledge, only VARSR (Qu et al., 2025), PURE (Wei et al., 2025), and the concurrent VARestorer (Zhu et al., 2026) have tackled ISR leveraging existing, powerful, AR models. PURE (Wei et al., 2025) uses a 7B Lumina-mGPT (Liu et al., 2024) multimodal backbone with an augmented image and text vocabulary where both an image and its described degradation are modelled together. On the other hand VARSR (Qu et al., 2025) and VARestorer (Zhu et al., 2026) build on VAR (Tian et al., 2024) for the ISR task. VARSR applies VAR directly using the LR as conditioning, whereas VARestorer tries to distill a pre-trained VAR into a one-step model. Both approaches inherit the limitations of VAR models (Tian et al., 2024) that cannot generate semantically meaningful images at the intermediate scales. Our proposed Hierarchical Tokenization strategy bypasses such problem while providing a strong inductive bias for the VAR training, resulting in much smaller models that deliver state of the art results.

**Direct Preference Optimization in ISR**   Direct Preference Optimization (DPO, (Rafailov et al., 2023)) offers a simple alternative to Reinforcement Learning from Human

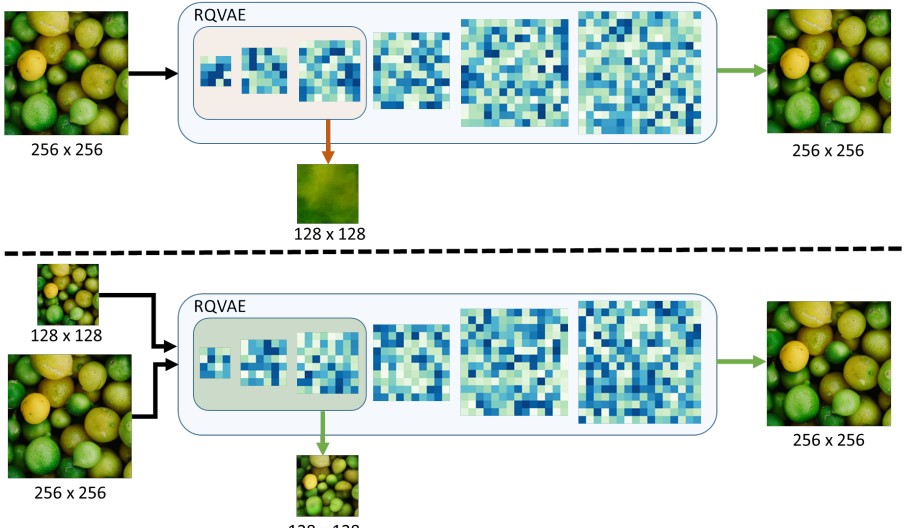

Figure 2: **Top**: we represent the quantization of an input image (up to $L = 6$ scales for clarity of visualization). We observe that reconstructing the image using all 6 residuals leads to perfect reconstruction. However, mapping the residuals up to the first three scales does not result in the desired $2\times$-downsampled version of the input image, i.e., *the initial scales do not convey scale-wise semantic information of the encoded image*. **Bottom**: Our proposed Hierarchical Image Tokenization; see Algorithm 1, tackles the quantization in a hierarchical manner, enforcing the first set of tokens to be shared across both resolutions, thereby allowing reconstruction at both scales.

Feedback (RLHF) that bypasses the need of a reward model, provided a set of preference pairs is given. Originally conceived for LLMs, DPO has gained attraction in the image generation domain using diffusion models (Wallace et al., 2024; Na et al., 2025), as well as in the domain of ISR (Wu et al., 2026; Yi et al., 2026; Zhu et al., 2025a). However, these methods rely on using an external reward model to generate a set of preference pairs for DPO. In this paper, we propose a DPO-based regularization term that is seamlessly integrated in the main training objective, exploiting the model's capacity to provide the log-likelihood of both the HR and LR images, something not feasible in diffusion-based approaches. Our DPO-based regularization helps the model favor HR outputs against simple bilinear, blurry interpolations of the input LR images.

## 3. Method

### 3.1. Background

**Image tokenization** Auto-regressive image generation extends standard LLM-based text prediction to the image domain, by replacing text tokens by image tokens. Akin to the former, image tokens are a set of embeddings forming a fixed, discrete vocabulary. Representing an image as a series of tokens is commonly known as *image tokenization*.

An image is *tokenized*, i.e. mapped to a latent representation, using an autoencoder endowed with a discrete quantizer that

maps continuous features to their closest representations from a fixed, learned vocabulary. The tokenizer is composed of an encoder $\mathcal{E}$, a vocabulary $\mathcal{V}$, and a decoder $\mathcal{D}$. The most common tokenizer is Vector Quantization (*VQ*) (Van Den Oord et al., 2017; Esser et al., 2021)), which maps the continuous embeddings given by the image encoder to their closest tokens in the vocabulary. Given an image $I \in \mathbb{R}^{3 \times H \times W}$, the encoder produces a latent representation $\mathcal{E}(I) = \mathbf{Z} \in \mathbb{R}^{n_z \times h \times w}$, composed of $h \times w$ continuous vectors $\mathbf{z}_{i,j} \in \mathbb{R}^{n_z}$. The latent resolution $h \times w = fH \times fW$ depends on the encoder compression ratio $f \leq 1$. Each of the latent vectors is then mapped to their closest representation in the codebook $\mathcal{V} = \{\mathbf{r}_k\}_{k=1}^{K} \subset \mathbb{R}^{n_z}$, with $K$ the vocabulary size. The quantization $\mathcal{Q}$ of a feature vector $\mathbf{z}_k \in \mathbb{R}^{n_z}$ is described as

$$\mathbf{r}_k = \mathcal{Q}(\mathbf{z}_k) = \arg\min_{\mathbf{r}_l \in \mathcal{V}} \|\mathbf{r}_l - \mathbf{z}_k\|. \tag{1}$$

The decoder is then tasked with reconstructing the input image $I$ from the discrete tokens $\mathbf{R} \in \mathbb{R}^{n_z \times h \times w} = \{\mathbf{r}_k\}$, as $\tilde{I} = \mathcal{D}(\mathbf{R})$. The tokenizer, i.e. the encoder, vocabulary, and decoder, is learned using a combination of image-reconstruction losses (e.g. $\ell_1$, perceptual, GAN) and a commitment loss (Van Den Oord et al., 2017). The tokenization of an image is then the process of representing it as a sequence of indices $\{z_1, ..., z_{h \times w}\}$ corresponding to the discrete vectors $\{\mathbf{r}_1, ..., \mathbf{r}_{h \times w}\} = \mathcal{Q}(\mathcal{E}(\cdot))$. The encoder and tokenizer convert an image into a series of tokens that can be modeled by an LLM, and the vocabulary and decoder

are used to map the output of the LLM to the image space. However, the discretization step introduces a quantization error to the autoencoder, which can limit its reconstruction capabilities, and ultimately the quality of the generated images. While larger values of $K$ should produce better results, the training of such an autoencoder is unstable.

To enhance the quality of image tokenization, Residual Quantization (*RQ* (Lee et al., 2022)) modifies *VQ* by decomposing the latent $\mathbf{z}$ into a series of $L$ entries $\mathbf{r}_{l=1}^{L}$, such that $\sum_{l=1}^{L} \mathbf{r}_l \approx \mathbf{z}$, and $|\mathbf{z} - \sum_{l=1}^{L''} \mathbf{r}_l| \geq |\mathbf{z} - \sum_{l=1}^{L'} \mathbf{r}_l|$ for all $L'' < L' < L$. That is, *RQ* applies $\mathcal{Q}$ to the residuals $\delta_{L'} = \mathbf{z} - \sum_{l=1}^{L'} \mathbf{r}_l$, with $\mathbf{r}_1 = \mathcal{Q}(\mathbf{z})$, and $\mathbf{r}_l = \mathcal{Q}(\delta_l)$, such that the cumulative vectors approach the latent vector $\mathbf{z}$. Similar to *VQ*, *RQ* is applied to each pixel of a latent representation $\mathbf{Z} \in \mathbb{R}^{n_z \times h \times w}$. While *RQ* results in longer sequences for image tokenization, it allows a smaller vocabulary size, helping the generation of better image sequences.

The approaches above use a *raster* tokenization, i.e. they quantize latent vectors in a left-to-right, up to bottom approach. This is needed for the LLM to model images in a sequential manner (*next token prediction*). However, this formulation ignores image correlations that occur far from nearby pixels. To address this limitation, VAR (Tian et al., 2024) proposed to modify the above formulation by making the residual $\Delta_{L'} = \mathbf{Z} - \sum_{l=1}^{L'} \mathbf{R}_l$ at each level $L'$ be quantized using an increasing number of tokens $n_{L'}$. That is, each level $L'$ is quantized using a number of tokens $n_{L'} \leq h \times w$. To do so, the quantization is done on each spatial location of $\Delta_{L'}$ after being downsampled to the target resolution $h' \times w'$. The map $\mathbf{R}_{L'} \in \mathbb{R}^{n_z \times h' \times w'}$ is then upsampled back to the native resolution $h \times w$. Using this hierarchical decomposition, each level is now referred to as *scale*. During generation the whole set of tokens corresponding to a given scale is produced using full attention.

**AR-based image generation**. Early approaches for AR-based image generation directly adopted the next-token prediction paradigm standard for LLMs (Esser et al., 2021). The modeling of an image $I$ represented as a series of tokens $\{z_1, ..., z_{h \times w}\}$ is defined as $p(z) = \prod_{i=1}^{L} p(z_i | z_{<i}, c)$, with $c$ a conditioning signal (e.g. a prompt). However, the recent approach of VAR shifted this paradigm to a next-scale prediction: an image is now represented as a series of discrete, ordered tokens $\{z_{i,j}\}$ with $i$ representing the scale and $j$ representing the position within the scale. The goal of the AR model is to represent the conditional probability of a sequence as:

$$p(z|c) = \prod_{i=1}^{L} p(\{z\}_i | \{z\}_{<i}, c), \qquad (2)$$

where $\{z\}_i = (z_{i,1}, \cdots z_{i,n_i})$ is the set of tokens corresponding to scale $i$.

### 3.2. Motivation

The next-scale prediction of VAR aligns with the principles of image super resolution. This was identified by VARSR (Qu et al., 2025), that observed that VAR can be seamlessly adapted for the task of ISR by simply setting the conditioning $c$ to a representation of the LR input image. In VARSR, the representation $c$ is directly that of a fine-tuned VQVAE. However despite the appealing properties of approaching ISR as an AR problem, we identify and aim to address the following two main shortcomings:

**a)** AR methods using a standard RQ-VAE fail to represent intermediate scales, as all residuals are needed to decode the final-scale image. Let $f \leq 1$ be the compression factor of the autoencoder. An input image of size $H \times W$ will produce a feature map $h \times w = fH \times fW$. In convolutional autoencoders, one can compress higher (or lower) resolution images using the same architecture whilst keeping $f$. In that sense, if an image is represented as a sequence of scales $\{z_{i,j}\}_{i=1,\cdots L, j=1 \cdots n_i}$, one would desire that the residuals up to $L' < L$ would lead the decoder to produce an image $H' \times W' = h_{L'}/f \times w_{L'}/f$. However, it is not guaranteed (and typically not true) that early residuals encode the semantic representation of the image, while additional scales add high-frequency details (see Figure 2 (top) for a visual example). This lack of guarantees means that if a model has been trained to produce $4\times$ upsample, it is not expected that the output at the scale corresponding to $2\times$ will indeed correspond to the $2\times$-upsampled image. We address this shortcoming with our **Hierarchical Image Tokenization**.

**b)** In addition to not being able to produce reliable intermediate results, we observe that VARSR, contrary to most extant ISR work, needs additional annotated images (i.e., positive and negative samples) to use in Image-based Classifier-Free Guidance to help approach the desired HR targets. We address this shortcoming with our **DPO-based regularization** loss, which results in superior performance while only using standard ISR training datasets.

### 3.3. Hierarchical Image Tokenization

To address point **a)** we propose to partition the scale-based quantization by progressively encoding increasingly larger resolutions of the input image. We start by defining $s \leq 1$ as the target scale measured with respect to the feature dimensions $h \times w$ of the image to be quantized, $I \in \mathbb{R}^{3 \times H \times W}$. In other words, while, $f$, in $h, w = fHf, fW$ is a fixed ratio, $s$ is the target scale in the feature space. We further define $\rho_L = (h, w)$ as the resolution of the feature map computed at the native resolution $H \times W$. We use the subscript $L$ to match the number of residual levels with the feature dimensions at the native resolution. Any further upscaling of the latent map would result in higher-resolution images. We

define $\mathcal{E}(I) = \mathbf{Z} \in \mathbb{R}^{n_z \times h \times w}$ as the corresponding image features, and $\mathbf{Z}_s \in \mathbb{R}^{n_z \times sh \times sw}$ as the features corresponding to the downsampled image $I_s \in \mathbb{R}^{3 \times sH \times sW}$. Starting from the smallest scale, we propose to tokenize the residuals for each scale until the dimension of the residual surpasses the dimension of the corresponding features for that scale, i.e. until we reach step $i = \arg\max_k \rho_k \mid \rho_k < s_i\rho_L$. When moving into the next scale, the previous tokens are used to compute the residuals up to the previous scale, and further quantization is performed for the current scale. In other words, we partition the resolutions for the residual quantization in non-overlapping subsets corresponding to the target scales. We then quantize the residuals as if the target scale was the last one, and append the tokens to the list computed from previous scales. The tokenization of scale $s_1$ sets the starting point for the tokenization of scale $s_2$, and so forth. Algorithm 1[1] summarizes the proposed tokenization procedure (also illustrated in Figure 2 (bottom)). At the end of the tokenization, an image is represented by a sequence of tokens as:

$$z = \left\{ \left\{ \underbrace{\{z_1, z_2, \ldots\}}_{s_1}, \ldots, z_l, z_{l+1}, \ldots \right\}, \ldots, z_L \right\} \quad (3)$$

where $s_i$ shows the set of tokens that can be used to decode scale $i$. This new image tokenization, based on downsampled versions of the image itself, will then be used to prepare the ground-truth sequences for the VAR training. However, because current quantizers do not have a vocabulary that is semantically consistent across scales, we finetune the vocabulary and the decoder of the RQ-VAE to accommodate the multi-scale quantization. To do this finetuning, we incorporate a scale-specific decoder, and compute the standard reconstruction losses for each scale. We then keep the decoder frozen and update the vocabulary using the gradient of the $\ell_2$ distance between the encoder features at each of the scales and the embeddings resulting from the tokenization.

Beyond the practical implications of Algorithm 1, we observe that our Hierarchical Image Tokenization serves as a **strong inductive bias** for the training of the model. By forcing intermediate scales to follow a predefined structure, the model is narrowly driven towards the best path for the residual quantization at the latest step. We observe empirically that such approach attains state of the art models with a much smaller model than VARSR, and without the need of collecting external data.

### 3.4. DPO regularization of VAR training

To address point **b)**, we observe that the model can be prone to bypass the difficulty of the ISR problem by directly pre-

---

**Algorithm 1** Hierarchical RQVAE Tokenization

**Inputs:** $\{I_n\}_{n=1}^N$
**Hyperparameters:** steps $L$, resolutions $(\rho_l)_{l=1}^L$, target scales $(0, s_1, \cdots, s_N)$
**for** $n = 1, \cdots, N$ **do**
  $\mathbf{Z}_n = \mathcal{E}(\text{interpolate}(I, s_n\rho_L))$
  **for** $i = 1, \cdots, \max_k \rho_k \leq s_i\rho_L$ **do**
    **if** $i < \max_k \rho_k \leq s_{i-1}\rho_L$ **then**
      $\mathbf{Z}_n = \mathbf{Z}_n - \phi(\text{interpolate}(\mathbf{R}_{n,i}, \rho_n))$[1]
    **else**
      $(z_{n,i}, \mathbf{R}_{n,i}) = \mathcal{Q}(\text{interpolate}(\mathbf{Z}_n, \rho_i))$
      $\mathbf{Z}_n = \mathbf{Z}_n - \phi(\text{interpolate}(\mathbf{R}_{n,i}, \rho_n))$
      $z \leftarrow [z; z_{n,i}]$
      $r \leftarrow [r; \mathbf{R}_{n,i}]$
**Return:** multi-scale tokens $z$

[1]$\phi$ is a convolutional layer introduced in (Tian et al., 2024) to address the information loss in the upscaling step.

---

dicting the LR tokens instead of the HR ones. This problem is also enhanced by the fact that there is an expected overlap between the tokens in both the LR and HR images. To overcome this, we enforce the model to not only maximize the log-likelihood of the ground-truth tokens, but also penalize the model when the output tokens are closer to describing a simple bilinear upsampling of the LR image. This type of objective is similar to that of Direct Preference Optimization (Rafailov et al., 2023), where the network is trained to prioritize a *preferred* sequence over the *non-preferred* one. Because we are not given a reference model, we simply choose a simpler regularization term of the form:

$$\mathcal{L}_{DPO} = -\log \sigma \left( \beta \log \frac{p(z_{HR})}{p(z_{LR})} \right). \quad (4)$$

While in (Rafailov et al., 2023) $\beta$ is a parameter controlling the deviation from the base reference policy, in our simplified formulation the value of $\beta$ in Equation (4) acts as an inverse, non-linear weighting factor for the loss. We follow prior work and set $\beta = 0.2$. We observed that too low values of $\beta$ make $\mathcal{L}_{DPO}$ almost constant, having no effect in the training. High values of $\beta$ caused training instability. To compute $p(z_{LR})$, we upsample the low-res degraded image to the largest scale and tokenize it following Algorithm 1. We define then $p(z_{LR}) = \prod_{z_i \in z_{LR}} (\exp o_{z_i} / (\sum_v \exp o_v))$ with $o = (o_1, \ldots, o_K)$ the output logits from VAR. To our knowledge, this is the first time a DPO-based regularization term is introduced in the context of AR image generation.

We train the VAR with our Hierarchical Image Tokenization approach to represent the ground truth. We use a combination of a cross-entropy loss and our proposed DPO regularization term with equal weights for the two losses[2]. To allow the VAR to handle multiple resolutions, we use an

---

[1]Please see Section A for a PyTorch-like pseudocode version.

[2]We report in Section B pseudo-code for DPO regularization

| Method | Metric | 128 | 256 | 512 |
|--------|--------|-----|-----|-----|
| Switti | PSNR↑ | - | - | 24.17 |
|        | LPIPs↓ | - | - | 0.115 |
| Switti* | PSNR↑ | 17.86 | 21.43 | 23.52 |
|         | LPIPs↓ | 0.285 | 0.159 | 0.144 |
| H-RQVAE | PSNR↑ | 20.10 | 22.34 | 24.81 |
|         | LPIPs↓ | 0.184 | 0.143 | 0.124 |

Table 1: Reconstruction results of different RQVAEs on ImageNet-512 (Deng et al., 2009) validation set. Switti∗ denotes results using Switti's checkpoint and the tokenization of Algorithm 1 *before* finetuning.

overparameterized learnable positional embedding which is downsampled to each of the $l$ resolutions $\rho_l$. Also, instead of using a ControlNet (Zhang et al., 2023) to encode the LR image as conditioning to the VAR, as in VARSR, we directly use the RQVAE encoder features. We bilinearly upsample the LR to the target resolution before encoding it, and use the corresponding features as conditioning.

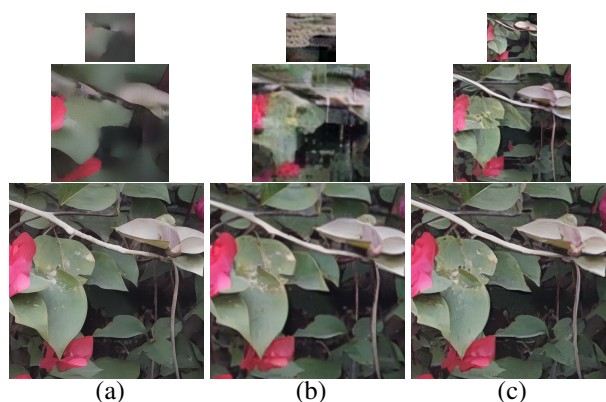

(a) (b) (c)

Figure 3: Multi-scale SR evaluation of (a) VARSR, (b) Baseline, and (c) Our proposed approach. The baseline is trained using our RQVAE, but by generating the ground-truth sequences without hierarchical tokenization. Top-to-bottom images correspond to output of the models at scales $128\times128$, $256\times256$, $512\times512$, respectively, corresponding to scale factors of $\times1$, $\times2$ and $\times4$. For VARSR we represent outputs at $144$ and $288$, respectively, which are the closest to the target outputs using their sequence of scales $(1, 2, 3, 4, 6, 9, 13, 18, 24, 32)$. Better viewed with zoom.

### 3.5. Implementation details

We follow prior work and use the standard $4\times$ setting where the input LR images are of $128 \times 128$ resolution, while the target is $512 \times 512$. We use three scales $s = (0.25, 0.5, 1)$, corresponding to $1\times$, $2\times$ and $4\times$ upsampling, respectively. The first scale corresponds to simple image denoising.

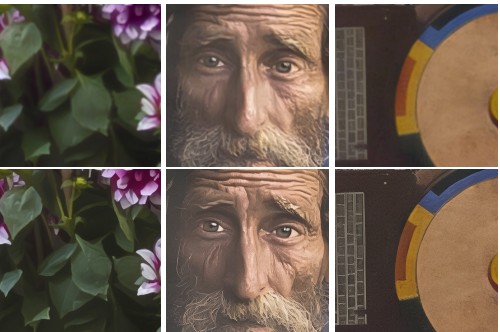

Figure 4: Qualitative results: (Top) **without** and (Bottom) **with** our proposed DPO-based regularization. We can see the role of the regularization term in sharpening results.

**Hierarchical RQVAE.** (H-RQVAE) The compression factor of the encoder is $f = 1/16$. To align with prior work on VAR, we use $L = 10$ steps, with resolutions $\rho_l = (4, 6, 8, 10, 14, 16, 20, 24, 28, 32)$. Steps $l = 1 \dots 3$ are mapped to the $128 \times 128$ output; steps $l = 1 \dots 6$ result in the $256 \times 256$ outputs, and steps $l = 1 \dots 10$ are used to perform the full $4\times$ upscale. Our RQVAE is initialized from the Switti checkpoint (Voronov et al., 2025). To finetune the RQVAE we follow a hybrid approach akin to that of HART (Tang et al., 2025), i.e., we randomly drop the quantization step and directly pass the encoder features to the decoder, with $50\%$ probability. We finetune our RQVAE vocabulary and decoders on the OpenImages (Kuznetsova et al., 2020) dataset for 25K iterations using AdamW (Loshchilov & Hutter, 2019) optimizer with batch size 384, a learning rate of 0.00025, and weight decay 0.05. The loss is defined as $\mathcal{L}_{RQVAE} = \ell_2 + 5\,\mathcal{L}_{LPIPs}$, with $\mathcal{L}_{LPIPs}$ being the standard LPIPs loss (Zhang et al., 2018a). We train our RQVAE with 24 A100 GPUs, taking $\sim 24$ hours to complete.

**Hierarchical VAR.** (H-VAR) The transformer follows the standard GPT-2 style (Radford et al., 2019), adopted by VAR (Tian et al., 2024) and VARSR (Qu et al., 2025). Contrary to VARSR, we use a transformer with only 16 blocks, resulting in a 310M parameter transformer. Following VARSR, we initialize our model from the VAR d-16 official checkpoint. The model is trained using 24 A100 GPUs for 200 epochs, with a batch size of 384, learning rate of 1e-3, weight decay 0.005, and using an AdamW optimizer with betas $(0.9, 0.95)$. We follow prior work and set $\beta = 0.2$ for the DPO-loss. It takes $\sim 13$ hours for the training to be completed. To compute the LR features, we upsample the LR image to $512 \times 512$ and encode it using the RQVAE encoder, producing a set of 1024 conditioning tokens. The total number of tokens is $\sum_l \rho_l^2 = 3452$. At inference, we use KV-cache for computational saving (Qu et al., 2025; Tian et al., 2024; Tang et al., 2025).

| Dataset | Metrics | 128 | | | 256 | | | 512 | | |
|---------|---------|-------|----------|-------|-------|----------|-------|-------|----------|-------|
| | | VARSR | Baseline | **H-VAR** | VARSR | Baseline | **Ours** | VARSR | Baseline | **H-VAR** |
| *RealSR* | PSNR↑ | 18.97 | 17.08 | **22.09** | 20.07 | 18.62 | **24.41** | 24.61 | **26.11** | 25.55 |
| | LPIPS↓ | 0.618 | 0.686 | **0.199** | 0.450 | 0.491 | **0.236** | 0.350 | 0.311 | **0.256** |
| *DRealSR* | PSNR↑ | 21.56 | 19.97 | **25.26** | 22.96 | 22.10 | **27.65** | 28.16 | 28.04 | **28.73** |
| | LPIPS↓ | 0.565 | 0.636 | **0.187** | 0.435 | 0.456 | **0.244** | 0.354 | 0.322 | **0.259** |

Table 2: Comparison with VAR-based baselines on the multi-scale ISR task; baselines clearly fail at intermediate scales.

| Dataset | DPO | Metric | 128 | 256 | 512 |
|---------|-----|--------|------|------|------|
| RealSR | ✗ | PSNR↑ | 20.56 | 23.09 | 25.72 |
| | | LPIPs↓ | 0.347 | 0.309 | 0.310 |
| | ✓ | PSNR↑ | 22.09 | 24.41 | 25.55 |
| | | LPIPs↓ | 0.1996 | 0.2357 | 0.256 |
| DRealSR | ✗ | PSNR↑ | 23.03 | 26.38 | 28.61 |
| | | LPIPs↓ | 0.311 | 0.311 | 0.335 |
| | ✓ | PSNR↑ | 25.26 | 27.65 | 28.73 |
| | | LPIPs↓ | 0.1871 | 0.2435 | 0.259 |

Table 3: Quantitative ablation of the DPO regularization.

# 4. Experiments

**Training datasets.** We adopt standard training datasets used for the ×4 ISR task and use a combination of DIV2K (Agustsson & Timofte, 2017), DIV8K(Gu et al., 2019), Flickr2k (Timofte et al., 2017), OST (Wang et al., 2018) and a subset of 10K images from FFHQ training set (Karras et al., 2019). We generate the LR-HR training pairs, using Real-ESRGAN (Wang et al., 2021) degradations.

**Testing datasets.** For evaluation, we follow recent ISR work, e.g., (Wang et al., 2024; Hu et al., 2025) and test on one synthetic dataset and two real ones. Specifically, we use the synthetic dataset made of 3K LR-HR ($128 \rightarrow 512$) pairs synthesized from the DIV2K validation set using the Real-ESRGAN degradation pipeline. For the real datasets, we use $128 \times 128$ center crops from the RealSR (Cai et al., 2019), DRealSR (Wei et al., 2020) datasets.

**Baselines.** We compare to GAN-based methods, diffusion-based methods, and also the closely related recent VARSR work. Importantly, given that VARSR relies on a big AR model (VAR-d24) and was trained on a much larger dataset that is not publicly available, we also retrain it using the same backbone model adopted in our work (VAR-d16) and using the same training data, for fair comparison.

**Evaluation metrics.** We evaluate using standard ISR metrics, including PSNR, SSIM, LPIPS, FID and MUSIQ.

## 4.1. Ablation study

We first provide support for each component in our framework with a thorough ablation study and then present extensive quantitative and qualitative comparisons in Sec. 4.2.

**RQVAE reconstruction.** We first evaluate the RQVAE reconstruction capabilities at different scales. We compare our finetuned RQVAE to the initial checkpoint without (i.e., Switti) and with (i.e., Switti*) our tokenization algorithm described in Algorithm 1. We show in Table 1 that our finetuned RQVAE surpasses alternatives in all metrics. While a small and expected drop occurs at higher scales due to the hierarchical nature (see discussion in the limitations section), we can see that our tokenization algorithm allows for decoding all scales with high fidelity.

**ISR at different scales.** Next, we evaluate the capacity of our proposed model to upsample images at different scales. In Table 2 we compare the results for VARSR against our proposed pipeline for 128, 256 and 512 resolution. To further showcase the importance of our proposed tokenization scheme for the VAR, we train a baseline VAR model using the same RQVAE we used, but *without the proposed hierarchical tokenization during VAR training*. We show in Tab. 2 and Figure 3 how our method is the only one capable of reconstructing the intermediate scales properly.

**Role of DPO regularization.** Finally, we evaluate the contribution of our proposed regularization term. We show numerically in Table 3, and qualitatively in Figure 4, that regularization helps the model predict sequences that align better with high resolution images, yielding sharper results.

## 4.2. Comparison with SOTA

**Results** We compare our proposed approach against the original VARSR and the retrained variant using the same model size and training datasets as ours. We also, compare to other GAN-based (BSRGAN (Zhang et al., 2021), Real-ESRGAN (Wang et al., 2021), SwinIR (Liang et al., 2021)) and Diffusion-based (LDM (Rombach et al., 2022), StableSR (Wang et al., 2024), ResShift (Yue et al., 2023)) ISR methods. The **quantitative results** shown in Table 4 and

| Dataset | Metrics | GAN-based | | | Diffusion-based | | | | | AR-based | | |
|---------|---------|-----------|--|--|------------------|--|--|--|--|-----------|--|--|
| | | BSRGAN | Real-ESR | SwinIR | LDM | StableSR | DiffBIR | SeeSR | ResShift | VARSR | VARSR-d16 | **H-VAR** |
| *DIV2K-Val* | PSNR↑ | 24.42 | 24.30 | 23.77 | 21.66 | 23.26 | 23.49 | 23.56 | 21.75 | 23.91 | 23.14 | 23.84 |
| | SSIM↑ | 0.616 | 0.632 | 0.619 | 0.475 | 0.567 | 0.557 | 0.598 | 0.542 | 0.598 | 0.579 | 0.601 |
| | LPIPS↓ | 0.351 | 0.327 | 0.391 | 0.489 | 0.323 | 0.364 | 0.328 | 0.428 | 0.326 | 0.495 | 0.317 |
| | FID↓ | 50.99 | 44.34 | 44.45 | 55.04 | 28.32 | 34.55 | 28.89 | 30.79 | 35.51 | 45.96 | 28.86 |
| | MUSIQ↑ | 60.18 | 59.76 | 57.21 | 57.46 | 65.19 | 65.57 | 68.35 | 70.02 | 71.27 | 50.90 | 69.40 |
| *RealSR* | PSNR↑ | 26.38 | 25.68 | 25.88 | 25.66 | 24.69 | 24.94 | 25.31 | 26.31 | 24.61 | 23.79 | 25.55 |
| | SSIM↑ | 0.765 | 0.761 | 0.767 | 0.693 | 0.709 | 0.666 | 0.728 | 0.742 | 0.717 | 0.647 | 0.726 |
| | LPIPS↓ | 0.266 | 0.271 | 0.261 | 0.337 | 0.300 | 0.349 | 0.299 | 0.346 | 0.350 | 0.413 | 0.256 |
| | FID↓ | 141.3 | 135.1 | 132.8 | 133.3 | 131.7 | 127.6 | 126.2 | 127.7 | 137.6 | 231.1 | 130.4 |
| | MUSIQ↑ | 63.28 | 60.37 | 59.28 | 56.32 | 65.25 | 64.32 | 69.56 | 69.17 | 71.26 | 53.70 | 69.58 |
| *DRealSR* | PSNR↑ | 28.70 | 28.61 | 28.20 | 27.78 | 27.87 | 26.57 | 28.13 | 28.46 | 28.16 | 27.30 | 28.73 |
| | SSIM↑ | 0.803 | 0.805 | 0.798 | 0.715 | 0.743 | 0.652 | 0.771 | 0.767 | 0.765 | 0.749 | 0.788 |
| | LPIPS↓ | 0.286 | 0.282 | 0.283 | 0.375 | 0.333 | 0.454 | 0.314 | 0.401 | 0.354 | 0.409 | 0.259 |
| | FID↓ | 155.6 | 147.7 | 146.4 | 164.9 | 148.2 | 160.7 | 147.0 | 159.8 | 155.9 | 244.7 | 145.1 |
| | MUSIQ↑ | 57.16 | 54.27 | 53.01 | 51.37 | 58.72 | 61.06 | 64.75 | 67.59 | 68.15 | 45.85 | 67.65 |

Table 4: Comparison to SOTA methods. Red and blue colors represent best and second-best results, respectively.

| $L$ | $\rho_l$ | Alloc. | **128** | **256** | **512** | Time (s) |
|-----|----------|--------|---------|---------|---------|----------|
| 10 | $(8, 8, 8, 16, 16, 16, 32, 32, 32, 32)$ | (3,3,4) | 23.4 (0.12) | 27.3 (0.09) | 31.5 (0.07) | 0.33 |
| | $(8, 8, 16, 16, 16, 16, 32, 32, 32, 32)$ | (2,4,4) | 21.2 (0.18) | 28.2 (0.08) | 31.5 (0.07) | 0.34 |
| | $(4, 6, 8, 10, 14, 16, 20, 24, 28, 32)$ | (3,3,4) | 23.4 (0.15) | 26.6 (0.11) | 31.4 (0.08) | 0.25 |
| 11 | $(4, 6, 8, 10, 14, 16, 20, 24, 28, 32)$ | (3,4,5) | 23.4 (0.15) | 26.6 (0.11) | 32.2 (0.06) | 0.31 |
| | $(8, 8, 8, 16, 16, 16, 32, 32, 32, 32, 32)$ | (3,4,5) | 23.4 (0.12) | 27.2 (0.09) | 32.3 (0.06) | 0.38 |

Table 5: Reconstruction metrics on DRealSR PSNR (LPIPS), across different allocations and resolutions, for the finetuned RQVAE using our proposed image tokenization. We highlight our chosen scheme in light grey. We observe that, as expected, metrics improve with $L = 11$, however resulting in a big jump in complexity. The time corresponds to the inference time for a corresponding VAR using a given scheme

| Model | Params | FLOPS | Time | DIV2K-Val |
|-------|--------|-------|------|-----------|
| Ours | 310M | 0.921T | 0.25s | 28.86 (0.317) |
| VARSR | 1B | 3.071T | 0.93s | 35.51 (0.326) |
| ResShift | 173M | 2.651T | 0.17s | 30.79 (0.428) |
| StableSR | 919M | 79.94T | 5.51s | 28.32 (0.323) |

Table 6: Complexity analysis. Along with parameters, FLOPS, and average inference time, we report the reconstruction metrics of each model on DIV2K-Val FID (LPIPS)

**qualitative comparison** shown in Figure 5 clearly demonstrate the effectiveness of our approach. Additionally, we report the computational complexity of our model against VARSR and the state-of-the-art diffusion-based models in Table 6. Notably, ResShift is the closest diffusion-based model in size to ours with $\sim 200M$ params, and VARSR-d16 is the most fair comparison to ours in terms of approach, model size, and training data. As shown in the second-to-last column of Table 4, VARSR-d16 without our HIT contribution is significantly worse across all datasets and metrics. This highlights that our hierarchical tokenization is not just

a multi-scale feature, but provides a strong inductive bias for the model. By enforcing semantically meaningful representation across scales, our method remains competitive against 1B-parameter models trained on massive, proprietary datasets, despite our model being a third of the size (310M), and trained entirely on small, standard SR datasets.

**Limitations.** By partitioning the number of steps into increasing scales, we are forcing the first levels to encode all information regarding scale $s_1$. This leaves a smaller number of steps to encode the second scale, and so forth. A full sequence of $L$ steps dedicated to representing a single resolution $H \times W$ is by definition stronger than if the same number of steps is required to encode the feature maps at lower resolutions as well. For example, if $L = 10$ and we use the first five steps to encode scale, $s_1 = 0.5$, the reconstruction at $H' \times W' = H/2 \times W/2$ will be better than that of existing RQVAEs, but will incur a small image degradation at the full resolution $H \times W$. Therefore, there is a tradeoff between the number of scales encoded by the sequence $L$ and the quality of the decoded images. In this paper, we follow prior work and use $L = 10$, and we partition the number of steps into three scales $s = (0.25, 0.5, 1)$.

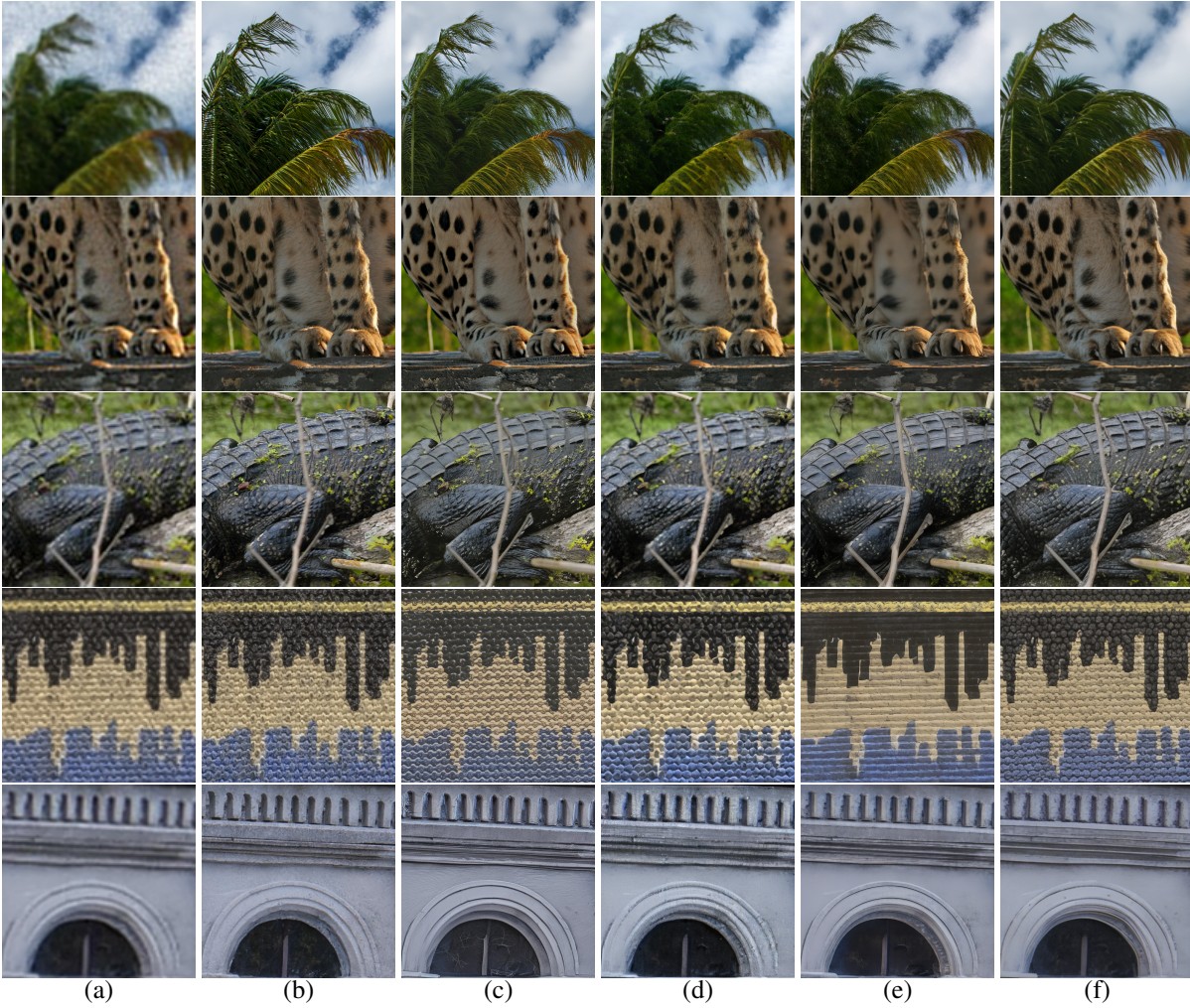

|  (a) | (b) | (c) | (d) | (e) | (f) |

Figure 5: Qualitative results. (a) Input LR (upsampled to target resolution); (b) Ground truth; (c) StableSR; (d) Resshift; (e) VARSR; (f) H-VAR. Zoom in for better view. Additional results can be found in Section C.

Increasing $L$ can however minimize the image degradation, albeit at the expense of a higher computational budget. To study the effect of varying $\rho_l$, $L$, and the partition scheme, we report in Table 5 the VQVAE reconstruction capacity at the three different scales, as well as the time that a VAR trained using such scheme would take to perform inference. We observe that increasing the number of levels in one comes at a high computational cost without a dramatic improvement in the reconstruction results.

## 5. Conclusion

In this paper, we advance the newly proposed paradigm of applying VAR to ISR. By introducing a hierarchical quantization approach, we enable VAR to decode multiple resolutions while performing the next-scale prediction, leading to semantically aligned intermediate results. Our proposed HIT scheme serves as a strong inductive bias for the model,

demonstrating the preference for such tokenization even for the target of fixed-scale super resolution. Importantly, we tackle multi-scale ISR using a simple yet effective training strategy with a DPO-based regularization term. Our model, trained on small standard datasets, attains state-of-the-art results with a reasonable 310M parameter transformer.

## Impact Statement

This paper aims at advancing the field of Image Super Resolution. There are many potential societal consequences of our work, none of which we feel must be specifically highlighted here.

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

## A. PyTorch-like pseudo-code for HIT

```python
import torch
import torch.nn.functional as F

def multires_to_idxBl(
        f_BChw_list, quantizer, patch_nums
    ):
    '''
    f_BChw_list: list of features at different resolutions i.e.
                 [E(I_0) , E(I_1), E(I_2)]
                 each tensor in the list is of increasing resolution
    quantizer: vocabulary and phi
    patch_nums: per-step resolution
    '''
    f_hats = []
    initial_idx = []
    for f_BChw in f_BChw_list: ## we start from lowest to highest res
        B, C, H, W = f_BChw.shape
        f_no_grad = f_BChw.detach()
        f_rest = f_no_grad.clone()
        f_hat = torch.zeros_like(f_rest)
        idx_V = []
        for si, pn in enumerate(patch_nums): # from small to large
            if pn > H:
                break
            if si < len(initial_idx):
                idx_N = initial_idx[si]
            else:
                # find the nearest embedding
                rest_NC = F.interpolate(f_rest, size=(pn, pn), mode='area')
                            .permute(0, 2, 3, 1)
                            .reshape(-1, C)
                rest_NC = F.normalize(rest_NC, dim=-1)
                idx_N = torch.argmax(
                    rest_NC @ F.normalize(quantizer.embedding.weight.data.T, dim=0), dim=1)

            idx_Bhw = idx_N.view(B, pn, pn)
            idx_V.append(idx_Bhw)
            h_BChw = F.interpolate(quantizer.embedding(idx_Bhw).permute(0, 3, 1, 2),
                                    size=(H, W), mode='bicubic').contiguous()
            h_BChw = quantizer.phi(h_BChw)
            f_hat = f_hat + h_BChw
            f_rest -= h_BChw

        initial_idx = idx_V
    return idx_V
```

## B. PyTorch-like pseudo-code for computing the DPO-loss

```python
import torchvision.transforms.resize as resize
import torch.nn.functional as F

def DPO_loss(
    ENCODER, VAR_MODEL,
    GT_img, LR_img,
    beta = 0.2
):
    '''
     ENCODER is used to compute the input features from a given lr or gt image
     VAR_MODEL is the trainable transformer
     GT_img is the ground-truth image
     LR_img is the low-res image
     beta is the DPO-loss temperature
    '''
    # Get ground-truth image tokenization
    gts = [resize(GT_img, res) for res in (128,256,512)]
    gt_features = [ENCODER(gt) for gt in gts]
    gt_idx_Bl = multires_to_idxBl(gt_features)
    gt_idx_Bl = [i.flatten(1) for i in gt_idx_Bl]

    # Get low-res image tokenization
    lrs = [resize(LR_img, res) for res in (128,256,512)]
    lr_features = [ENCODER(lr) for lr in lrs]
    lr_idx_Bl = multires_to_idxBl(lr_features)
    lr_idx_Bl = [i.flatten(1) for i in lr_idx_Bl]

    # Get transformer logits
    logits_BLV = VAR_MODEL( ... )

    # Compute loss
    N = logits_BLV.shape[1]
    pref = torch.cat(gt_idx_Bl,dim=1)[:,:N]
    nopref = torch.cat(lr_idx_Bl, dim=1)[:,:N]
    pref_logits = logits_BLV[:,:N,:].gather(dim=-1, index=pref.unsqueeze(2)).squeeze()
    nopref_logits = logits_BLV[:,:N,:].gather(dim=-1, index=nopref.unsqueeze(2)).squeeze()
    loss = -F.logsigmoid(beta*(pref_logits - nopref_logits)).mean() * 0.5

return loss
```

## C. Additional Qualitative Results

We present in Fig. 6 some additional qualitative examples, in addition to the extensive Figure 5 presented in the main paper.

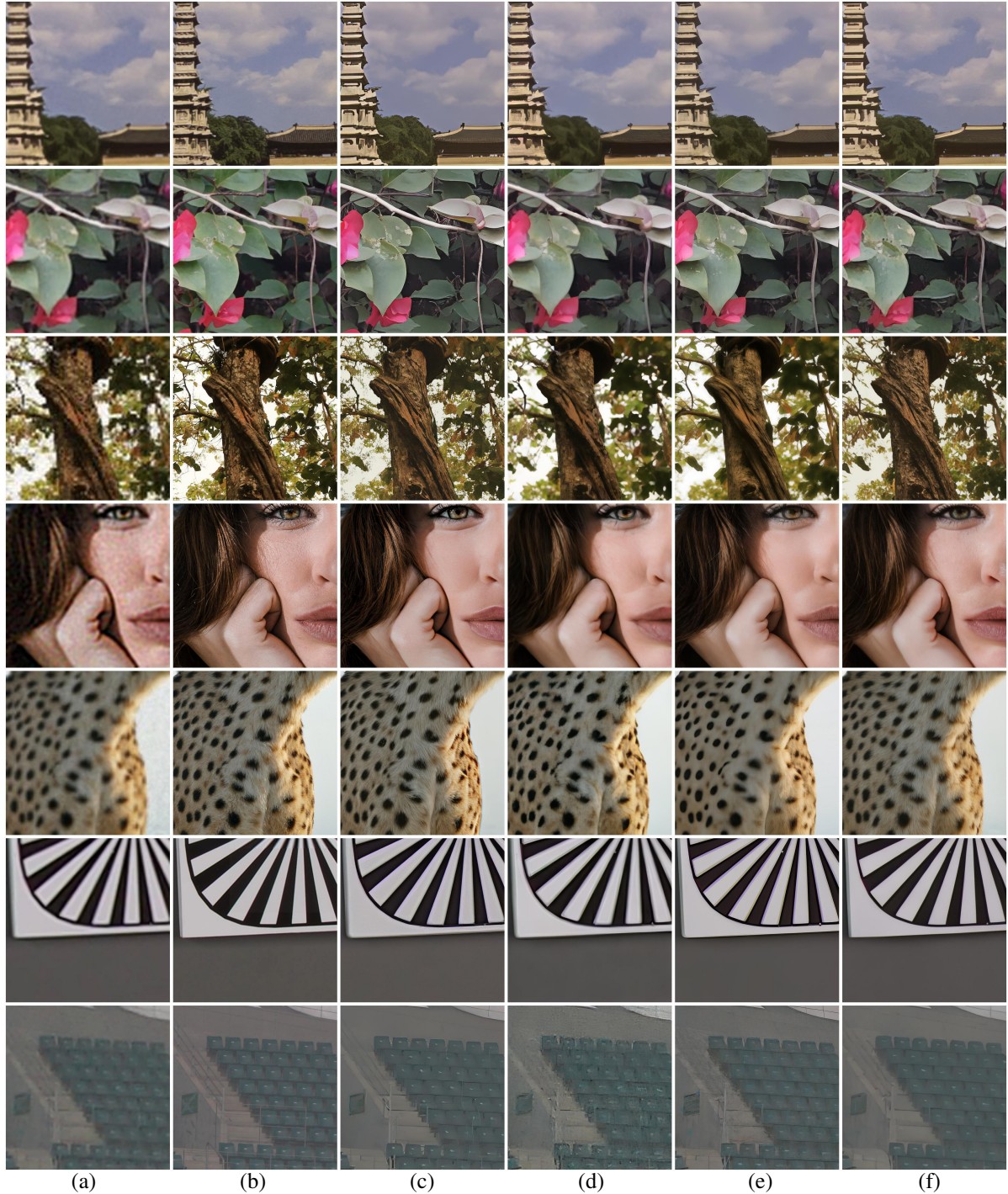

Figure 6: Qualitative results. (a) Input LR (upsampled to target resolution); (b) Ground truth; (c) StableSR; (d) Resshift; (e) VARSR; (f) Ours. Zoom in for better view.

