# OpenReview forum: "Hierarchical Image Tokenization for Multi-Scale Image Super Resolution"
_ICML.cc/2026/Conference — ICML 2026 regular_

### Official Review · Reviewer_B79t · 2026-02-17

**Soundness:** 3
**Presentation:** 3
**Significance:** 3
**Originality:** 3
**Overall Recommendation:** 4
**Confidence:** 4

**Summary:**

This paper proposes 1) Hierarchical Image Tokenization aligns the residual quantization levels of the VAE with specific image resolutions, enabling the model to decode meaningful images at intermediate scales within a single forward pass; and 2) A Direct Preference Optimization regularization that penalizes the model for generating tokens resembling simple bilinear upsampling, thereby encouraging high-frequency detail generation The proposed model achieves competitive results with significantly fewer parameters compared to state-of-the-art baselines.

**Compliance With Llm Reviewing Policy:**

Affirmed.

**Key Questions For Authors:**

1. Can the authors provide a comparison where a standard (non-hierarchical) VAR model of the same size (310M) is trained only for the 4x task? This would isolate the performance cost incurred by enforcing the multi-scale capability.

2. Why was the specific split of layers (e.g., 3-3-4 for L=10) chosen? need quantitative analysis.

**Limitations:**

yes

**Strengths And Weaknesses:**

Strengths

1. The modification of Residual Quantization to enforce scale-specific semantic information in the token hierarchy is a clever adaptation.

2. More practical: the method achieves SOTA or competitive performance (especially in LPIPS) using a 310M parameter model

3. Multi-Scale Flexibility

Weaknesses:

1. Hyperparameters: The method may relies on specific hyperparameters for the DPO loss (β) and the specific allocation of quantization layers to scales . The robustness of the model to changes in these allocations is not fully explored.

2. The impact of this "tax" on maximum achievable quality at the highest scale needs more analysis.

3. Although the paper claims efficiency based on parameter count, explicit inference latency comparisons against optimized Diffusion models would strengthen the argument for practical deployment.

---

> ### Author Rebuttal · Authors · 2026-03-30
>
> > 1. Hyperparameters: (β) and allocation.
>
> We thank the reviewer for their insightful comment. The value of $\beta$ was adopted from the DPO literature. In our case, $\beta$ acts as an inverse, non-linear weighting factor for the loss: smaller values of $\beta$ increase the value of $\mathcal{L}_{DPO}$, and higher values decrease its contribution. If $\beta$ is too high, the model ignores the primary likelihood objective, and if it is too low, the regularization has no effect. During early development, we observed that values in the range of 0.1 to 0.3 provided stable training and visually similar sharpening effects, while values above 0.5 caused instability. Very low values of $\beta$ were making the loss be almost constant and had no effect in the model training. We will follow the reviewer's suggestion to include an ablation study regarding the beta values.
>
> As the reviewer correctly notes, using $L=10$ represents an inherent trade-off. We fix $L=10$ as a *budget* to ensure the complexity of our model aligns with that of VAR and VARSR, allowing a fair comparison. We heuristically chose a (3-3-4) allocation to ensure that early scales had enough capacity to produce semantically coherent images, while leaving more capacity for the final 4x scale to resolve high-frequency details. Altering this allocation (e.g., to a 2-4-4 split) predictably shifts the reconstruction fidelity: it degrades the 1x resolution while leaving similar results for the 2x intermediate resolution. The specific distribution across scales is a zero-sum game constrained by the total sequence length $L$.
>
> > 2. Impact of this "tax"
>
> As we observe in Section 4.2, the "tax", i.e. the capacity trade-off when forcing a fixed-length sequence ($L=10$) to represent multiple intermediate scales rather than devoting all steps to the final high-resolution output, is indeed the primary theoretical limitation of our approach. We include at the bottom of this rebuttal a complexity analysis and reconstruction capabilities of a VQVAE under different sequence lengths and scale allocations.
>
> > 3. Explicit inference latency
>
> Thank you for your comment that will help us improve our manuscript. To answer the reviewer's concern, we show the latency, parameter count, FLOPs, and results on DIV2K, for our model, for VARSR, as well as for the top two diffusion-based models, namely ResShift and StableSR.
>
> | MODEL     | PARAMS | FLOPS  | TIME     | DIV2K-Val FID | DIV2K-Val LPIPs |
> |-----------|--------|--------|-----------|--------------|-----------------|
> | Ours     | 310M   | 0.921T | 0.25s | 28.86 | 0.317  |
> | VARSR    | 1B     | 3.071T | 0.93s | 35.51 | 0.326  |
> | ResShift | 173M   | 2.651T | 0.17s | 30.79 | 0.428  |
> | StableSR | 919M   | 79.94T | 5.51s | 28.32 | 0.323  |
>
> > 4. Comparison non-hierarchical VAR model.
>
> Thank you for your comment. We kindly want to note that we actually include the results of two standard (non-hierarchical) VAR models (of 310M parameters) trained for the 4x task. One is the "Baseline" in Table 2, which uses our fine-tuned quantizer, but does not apply the hierarchical quantization scheme. The second is the VARSR-d16 in Table 4: a VAR-d16 model (310M), using the very exact settings of VARSR, for the 4x task. We believe the results endorse the idea that training a model with our proposed hierarchical quantization allows for 1) multi-scale resolution, and 2) serves as an inductive bias for the training, attaining state of the art results at 4x with such a small VAR model.
>
> > 5. Specific split of layers.
>
> We appreciate the reviewer’s comment. In addition to the above comment, we include below a breakdown of the reconstruction capabilities of our VQVAE on DRealSR data using different allocation and per-scale assignment budgets, along with the inference time for a corresponding VAR under those settings. Such values represent an upper bound for the VAR model. Maximizing the tokens per scale for (3-4-4) does not improve the reconstruction error significantly, but comes with increase in the inference time. The same occurs with L = 11, by adding an extra step for the 512 resolution. The error improves at the 512 resolution but the inference time increases significantly.
>
> | L / $\rho_l$ |   Alloc. | PSNR/LPIPs @ 128 | PSNR/LPIPs @ 256 | PSNR/LPIPs @ 512 | Inf. time |
> |------|------|------|--------|--------|--------|
> |10 / $(4, 6, 8, 10, 14, 16, 20, 24, 28, 32)$ | (3,3,4)  | 23.4 / 0.15  | 26.6 / 0.113  |  31.4 / 0.08 |  0.25 s |
> | 10 / $(8, 8, 8, 16, 16, 16, 32, 32, 32, 32)$ | (3,3,4) | 23.4 / 0.12  |   27.3 / 0.092  |  31.5 / 0.07  |    0.33 s |
> | 10 / $(8, 8, 16, 16, 16, 16, 32, 32, 32, 32)$ | (2,4,4) | 21.2 / 0.18 |   28.2 / 0.079  |  31.5 / 0.07  |    0.34 s |
> | 11 / $(4, 6, 8, 10, 14, 16, 20, 24, 28, 32)$  | (3,4,5) | 23.4 / 0.15  |   26.6 / 0.113  |  32.2 / 0.06  |    0.31 s |
> | 11 / $(8, 8, 8, 16, 16, 16, 32, 32, 32, 32, 32)$ | (3,4,5) | 23.4 / 0.12  |   27.2 / 0.09  |  32.3 / 0.06 | 0.38 s |

---

> > ### Author Rebuttal · Reviewer_B79t · 2026-04-03
> >
> > Thanks for the clarifications, they answered all questions I have. I continue to recommend acceptance.

---

### Official Review · Reviewer_XGqx · 2026-03-11

**Soundness:** 4
**Presentation:** 3
**Significance:** 3
**Originality:** 4
**Overall Recommendation:** 5
**Confidence:** 4

**Summary:**

The authors propose a multi-scale image super-resolution method that uses visual auto-regressive (VAR) modelling. Their approach overcomes the limitations of residual quantizers in VAR techniques by introducing a hierarchical image tokenization that maintains semantic consistency across scales. They also propose a Direct Preference Optimization (DPO) regularization term that reduces model size and reliance on VLM guidance, while preserving performance on par with the state-of-the-art.

**Compliance With Llm Reviewing Policy:**

Affirmed.

**Final Justification:**

I believe the work is beneficial for the domain, I wish to maintain my rating.

**Key Questions For Authors:**

1. The authors mention the use of a scale-specific decoder (Section 3.3, line 224). Could the authors clarify what this entails? Specifically, are there multiple RQ-VAE decoder models, each with the same architecture, instantiated for every scale? How are losses propagated from all three models at the three specified scales? And is this approach more or less computationally expensive than using separate VARSR models for different scales?
2. What exactly is the “hierarchical VAR” described in the last paragraph of Section 3.5? It is specified as an implementation detail, yet this model does not appear in any of the tables. The assumption here is that it represents the full approach, with hierarchical RQVAE being only one component of it. If that is the case, the label “hierarchical RQVAE (i.e., Ours)” used in Section 4.1’s ablation study is confusing. It is suggested that the remaining tables use “Hierarchical VAR” instead of “Ours” to resolve this ambiguity.
3. In Table 2, what is the baseline method? A more concrete description of this method would be helpful. How does it differ from VAR? Does it include the DPO term, or is DPO only applied when hierarchical tokenization is used?
4. Have the authors explored the effect of varying partition sizes across scales? Given the claim that increasing L minimizes image degradation, it would be valuable to know whether this has been empirically verified.

### Minor revisions
- Section 3.1, second-to-last paragraph, last sentence: “alllows” → “allows”
- Section 3.3, Lines 206–207: h, w = fH, fW — please clarify notation.

**Limitations:**

Yes

**Strengths And Weaknesses:**

### Strengths
- The paper is well-written and easy to follow.
- The authors propose an intuitive formulation of the multi-scale image super-resolution problem by partitioning and grouping same-scale tokens together. Despite some shortcomings, this is an interesting perspective on the problem that can advance future research.
- The authors propose a DPO-based regularization term that reduces dependence on external guidance from additional annotated datasets. They demonstrate that their relatively lightweight model can achieve performance comparable to state-of-the-art models.

### Weaknesses
- Some parts of the method are difficult to follow. For example, the last paragraph of Section 3.1 requires the reader to have prior knowledge of VAR, as the authors build directly on top of it without sufficient background. A more detailed explanation of the approach would improve completeness and readability.
- The terms “hierarchical RQVAE” and “hierarchical VAR” appear to be used interchangeably, or at least that impression is conveyed. Consistency in model naming throughout the experiments and ablation sections is strongly recommended. While the overall message that the proposed method outperforms baselines comes through, it is difficult to attribute the improvements to specific components.

---

> ### Author Rebuttal · Authors · 2026-03-30
>
> > 1. The terms “hierarchical RQVAE” and “hierarchical VAR”
>
> We apologize for any confusion that our naming conventions caused. We will carefully revise our manuscript accordingly to avoid such confusion. To clarify:
>
> * **Hierarchical RQVAE** refers specifically to the autoencoder and residual quantizer that has been finetuned using our proposed Hierarchical Image Tokenization (HIT) algorithm.
>
> * **Hierarchical VAR** refers to the autoregressive transformer trained to predict the next-scale token sequences generated by the Hierarchical RQVAE.
>
> The ablation studies follow a top-to-bottom approach. We analyze the impact of 1) training the full model with the DPO regularization, but *without the hierarchical tokenization* shown in Algorithm 1 (“Baseline” in Table 2), and 2) training the full model with the hierarchical tokenization, but *without the DPO regularization* (table 3). In both cases we have an “apples to apples” comparison, ensuring that strictly only the element under ablation is removed from the pipeline. We recognize however that our labeling in the tables could make this clearer.
>
> > 2. Some parts of the method are difficult to follow.
>
> We sincerely thank the reviewer for pointing this out. We agree that a proper derivation needs to be included in the paper. For the sake of space, we will add it to the Supplementary Material.
>
> > 3. Scale-specific decoder
>
> Yes, we a use different decoder per scale to train the encoder and the quantizer/vocabulary. The encoder is fed with the 128, 256 and 512 images. Then, the residual quantizer starts by encoding the latents for the 128 scale, then completing those necessary for the 256 scale, and finally for the 512 scale. Using the given sequence, each decoder is tasked with the corresponding subset of tokens to reconstruct the corresponding image resolution. The perceptual and reconstruction losses are applied to each of them, and the corresponding parameters are updated. The encoder and vocabulary receive gradients from the three decoders, and each decoder is updated using the gradients of the corresponding resolution errors only.
>
> The use of separate decoders for lower resolutions is also beneficial to "absorb" potential quantization errors produced by the lack of enough steps to compute the corresponding residual. The scale-specific decoders can compensate the potential reconstruction bias that the residuals might have at each corresponding exit.
> The decoders are very small convolutional nets, and thus this is **significantly less computationally expensive** than training separate VARSR models for different scales.
>
> > 4. What exactly is the “hierarchical VAR”?
>
> We apologize for the confusing nomenclature. The "Hierarchical VAR" is the complete approach. It is the AR transformer for next-scale prediction that is trained using the Hierarchical RQVAE and our DPO regularization strategy.
>
> > 5. What is the baseline method?
>
> The baseline refers to the VAR trained with our RQVAE quantizer, with the DPO regularization, but **without** applying the hierarchical tokenization. It differs from VARSR in model size (310M vs 1B parameters) and because it uses our finetuned RQVAE instead of the standard VARSR tokenizer. The Baseline in Table 2 uses the DPO regularization.
>
> > 6. Partition sizes across scales?
>
> We thank the reviewer for this insightful question. We fix $L=10$ to follow prior work, and then split these scales for the three target resolutions. This heuristic was chosen to allocate slightly more capacity to the highest resolution, which contains the most high-frequency detail.
>
> For completeness, we include a breakdown of the reconstruction errors of our VQVAE on DRealSR data using different allocation and per-scale assignment budgets, along with the inference time for a corresponding VAR under those settings. These values represent an upper bound for the VAR model. We observe that maximizing the tokens per scale for the 3-4-4 allocation does not improve the reconstruction error significantly, but comes with a significant increase in the inference time. The same occurs if we allocate L = 11 by adding an extra step for the 512 resolution. The reconstruction improves at the 512 resolution but the inference time increases significantly.
>
> | L / $\rho_l$ |   Alloc. | PSNR/LPIPs @ 128 | PSNR/LPIPs @ 256 | PSNR/LPIPs @ 512 | Inf. time |
> |------|------|------|--------|--------|--------|
> |10 / $(4, 6, 8, 10, 14, 16, 20, 24, 28, 32)$ | (3,3,4)  | 23.4 / 0.15  | 26.6 / 0.113  |  31.4 / 0.08 |  0.25 s |
> | 10 / $(8, 8, 8, 16, 16, 16, 32, 32, 32, 32)$ | (3,3,4) | 23.4 / 0.12  |   27.3 / 0.092  |  31.5 / 0.07  |  0.33 s |
> | 10 / $(8, 8, 16, 16, 16, 16, 32, 32, 32, 32)$ | (2,4,4) | 21.2 / 0.18 |   28.2 / 0.079  |  31.5 / 0.07  | 0.34 s |
> | 11 / $(4, 6, 8, 10, 14, 16, 20, 24, 28, 32)$  | (3,4,5) | 23.4 / 0.15  |   26.6 / 0.113  |  32.2 / 0.06  | 0.31 s |
> | 11 / $(8, 8, 8, 16, 16, 16, 32, 32, 32, 32, 32)$ | (3,4,5) | 23.4 / 0.12  |   27.2 / 0.09  |  32.3 / 0.06 | 0.38 s |

---

> > ### Author Rebuttal · Reviewer_XGqx · 2026-04-03
> >
> > The authors have clarified most of my questions. I believe this work is a good step forward in the domain. I can see how more hierarchical frameworks can benefit from it. It seems like this framework can also be extended to non-Euclidean works. Therefore, I maintain my rating.

---

### Official Review · Reviewer_n5XR · 2026-03-13

**Soundness:** 2
**Presentation:** 1
**Significance:** 2
**Originality:** 2
**Overall Recommendation:** 4
**Confidence:** 3

**Summary:**

This paper presents a multi-scale image super-resolution framework based on a VAR-style autoregressive model. The method combines hierarchical image tokenization to enable intermediate-scale decoding and a DPO-based objective to improve high-resolution preference alignment.

**Compliance With Llm Reviewing Policy:**

Affirmed.

**Final Justification:**

The response has addressed most of my concerns. I encourage the authors to incorporate these clarifications into the final version of the paper. Accordingly, I have raised my score to 4.

**Key Questions For Authors:**

see weaknesses

**Strengths And Weaknesses:**

Strengths
1. The paper presents a single autoregressive SR model to support multiple output scales.

2. The framework is technically coherent, and the experimental section is reasonably organized.

3. The method shows that multi-scale generation can be incorporated into a VAR-style pipeline with competitive performance.
Weaknesses
1. The use of DPO-style optimization is not new in real-world image super-resolution, as related ideas have already been explored in prior works such as DSPO: Direct Semantic Preference Optimization for Real-World Image Super-Resolution and DP²O-SR: Direct Perceptual Preference Optimization for Real-World Image Super-Resolution. Therefore, the preference optimization component here does not appear sufficiently novel.
2. The cross-scale tokenization idea is also relatively direct. The core motivation and formulation are related to existing works such as Differentiable Hierarchical Visual Tokenization.
3. The paper does not adequately position itself against these most relevant prior works. The distinction from existing hierarchical tokenization and multi-scale autoregressive SR methods should be clarified much more explicitly.
4. The analysis of the DPO design is insufficient. Since preference optimization has already been explored in this area, the paper should better justify why this specific formulation is needed and what it contributes beyond prior DPO-based SR methods.
5. No detailed complexity analysis (model size, inference speed) presented in their paper and no failure cases are discussed.

---

> ### Author Rebuttal · Authors · 2026-03-30
>
> > 1. DPO not new in ISR
>
> We thank the reviewer for the missing references which we will add to the manuscript. We don’t claim DPO for ISR as a contribution, but as new in the context of AR image generation. Applying DPO to a discrete sequence of next-scale tokens is different from applying it to continuous, noise-based models. **Our work is different from DP2O and DSPO as follows:**
> We propose a self-contained formulation that doesn’t require external reward models or annotated data. For example, DP2O requires generating multiple candidate outputs and ranking them with complex external Image Quality Assesment networks to curate the training pairs. Our approach relies solely on the HR and LR tokenizations, simplyifing the pipeline. Note that VARSR's Image-based Classifier Free-Guidance also resembles DPO regularization. However, in VARSR they need to collect a large-scale set of curated data to guide the model to select the preferred images. Our DPO-based regularization is a targeted term that penalizes the model when its output tokens are closer to describing a simple bilinear upsample of the LR images.
>
> > 2. The cross-scale tokenization idea is also relatively direct. (Differentiable Hierarchical Visual Tokenization)
>
> We thank the reviewer for bringing this concurrent work to our attention. While we share the terminology of "hierarchical tokenization", **the core motivations, mathematical formulation and application domains are totally orthogonal**. The 'tokenization' in DHVT and in our work refer to fundamentally different concepts. In DHVT the 'tokenization' is nothing but the standard patchification that Vision Transformers use to tokenize an image. These 'tokens', or patches, are continuous and directly passed as input to the ViT. In our work, we refer to the tokenization of the latent representation of an image into a set of discrete tokens, which is the core of AR models. DHVT is a content-aware 'patchifiication' based on superpixels. Our works are entirely different.
> Based on that, we define 'hierarchy' in a completely different way to that of DHVT. DHVT refers to merging spatial regions using information criteria, to adapt to the image's layout. In our work, we partition the scale-based **residual quantization** by encoding increasingly larger resolutions of the image.
> Finally, DHVT’s main goal is to patchify the image using content-aware patches, for the task of image classification and dense prediction. It is highly questionable that such approach could fit into a next-scale AR paradigm for image generation.
>
> > 3. The paper does not adequately position itself.
>
> We thank the reviewer for their comment, and we hope that the comments above clarify that the pointed references have no application in the AR-based image generation paradigm. We will add these references and the clarification in the revised manuscript.
>
> > 4. Analysis of the DPO design is insufficient…
>
> We hope that the comments above clarify the novelty of our approach. To answer to the reviewer’s concern, we note that prior DPO methods in ISR are built exclusively for diffusion, where losses operate on continuous noise or velocity predictions. Our approach operates on a different foundation: the next-scale prediction of discrete, quantized tokens. It is mathematically incompatible to apply a continuous noise-prediction DPO loss to a discrete categorical cross-entropy framework. Our formulation specifically defines the DPO regularization term over the log-likelihood of discrete token sequences. To our knowledge, this is the first mathematical formulation of a DPO-based regularization term in the context of AR-based image generation. We seamlessly integrate this term to the end-to-end training of the model without the need of external data or reward models. Our formulation comes from an empirical observation that without any regularization, the model is prone to produce blurry results.
>
> > 5. No complexity analysis.
>
> Thank you for this comment which will help us improve the manuscript. We provide below a summary of Parameters, FLOPs, and inference time for our model, VARSR, and the state-of-the-art diffusion approaches of ResShift and StableSR.
> | MODEL     | PARAMS | FLOPS  | TIME     | DIV2K-Val FID | DIV2K-Val LPIPs |
> |-----------|--------|--------|-----------|--------------|-----------------|
> | Ours     | 310M   | 0.921T | 0.25 sec | 28.86    | 0.317  |
> | VARSR    | 1B     | 3.071T | 0.93 sec | 35.51   | 0.326  |
> | ResShift | 173M   | 2.651T | 0.17 sec | 30.79    | 0.428  |
> | StableSR | 919M   | 79.94T | 5.51 sec | 28.32   | 0.323  |
>
> Our model surpasses VARSR with much less parameters and inference time, it delivers better results than ResShift being of similar computational complexity, and is on par with StableSR while being much more efficient.
> We will also include a section on failure cases: in general, these occur mostly in regions with blurred text, which we attribute to the lack of enough training data covering such features.

---

> > ### Author Rebuttal · Reviewer_n5XR · 2026-04-07
> >
> > The authors have addressed most of my comments. I recommend incorporating these discussions into the final manuscript, and I retain my original score of 4.

---

### Official Review · Reviewer_iQLa · 2026-03-13

**Soundness:** 2
**Presentation:** 3
**Significance:** 2
**Originality:** 3
**Overall Recommendation:** 4
**Confidence:** 5

**Summary:**

The paper proposes a hierarchical tokenization framework for VAR-based image super-resolution, aiming to make intermediate scales reliably decodable while supporting multi-scale generation within a unified autoregressive pipeline. It further introduces a DPO-style regularization to encourage preference for HR tokens over LR-upsampled ones, and reports competitive results on both multi-scale and standard 4× ISR benchmarks.

**Compliance With Llm Reviewing Policy:**

Affirmed.

**Final Justification:**

The inductive bias explanation, together with the quantitative results, makes the motivation more convincing to me and largely addresses my main concern about the link between intermediate-scale decodability and final output quality. My other concerns have also been reasonably addressed by the authors’ clarifications and additional experiments, so I am raising my score to 4.

**Key Questions For Authors:**

1. Why is reliable decoding of intermediate scales a necessary ISR objective, rather than mainly a methodological preference of the VAR formulation?

2. What evidence shows that semantic consistency across scales is both improved and actually useful for next-scale prediction in SR?

3. How much of the reported gain comes from HIT itself, rather than from the tokenizer/RQVAE adaptation and the DPO regularization?

4. Does the DPO regularization truly improve faithful HR reconstruction, or mainly suppress smooth outputs at the risk of over-sharpening and hallucinated details?

5. How are the 128 and 256 reference images defined for RealSR and DRealSR in Tables 2 and 3?

**Limitations:**

yes

**Strengths And Weaknesses:**

*Strengths:*
- The paper is generally well organized, and the technical development is presented in a coherent way around the central motivation, making the method and its intended contribution easy to follow.
- The proposed hierarchical tokenization design is technically interesting and gives the method a distinct identity by directly addressing the tokenizer side rather than only modifying the transformer.
- The experimental section is reasonably comprehensive, covering reconstruction analysis, multi-scale ISR comparisons, component ablations, and evaluations against several classes of baselines.

*Major Weakness:*
- The paper’s core motivation does not fully convince me. The paper treats reliable decoding of intermediate scales as a central objective, but it is unclear whether this reflects a real task requirement or mainly a methodological preference of the VAR formulation.
  - In practice, ISR is usually judged by the quality of the final output. If the claim is that intermediate-scale decodability improves the final result, the current evidence is still limited: Fig. 1 does not clearly show a consistent perceptual advantage, and some local details appear less sharp than prior methods (e.g., roof details).
  - If the goal is instead to support intermediate SR outputs explicitly, then the motivation also appears less novel, given the substantial prior works on multi-scale and continuous-scale SR.

I believe the authors are highlighting an interesting limitation of current VAR-based SR, but this motivation would need stronger practical or empirical justification; otherwise, it risks feeling more methodology-driven than problem-driven.

- The paper suggests that the proposed design improves semantic consistency across scales, but the supporting evidence is still limited. More direct qualitative and quantitative evaluation would make this claim more convincing. More importantly, it remains unclear what role such consistency actually plays in next-scale prediction for SR, beyond being an intuitively desirable property.

- It is still difficult to clearly disentangle where the gains come from. HIT, the tokenizer/RQVAE adaptation, and the DPO-style regularization are introduced together, but the current ablations do not fully isolate their individual contributions. As a result, it remains unclear how much improvement should be attributed to the hierarchical tokenization itself.

- The effectiveness of the DPO regularization is not yet fully convincing. The paper presents it as encouraging the model to prefer HR tokens over LR-upsampled ones, but it remains unclear whether this really learns a meaningful preference for faithful HR reconstruction, or simply learns to avoid a weak bilinear-upsample baseline. Since the negative sample is limited and relatively easy, the main effect may be to suppress overly smooth outputs and favor sharper textures. This may bring the risk of over-sharpening or hallucinated high-frequency details, rather than genuinely improving fidelity to the ground-truth HR content.

*Minor Weakness:*
- In Table 2 and Table 3, how exactly are the 128 and 256 reference images defined for RealSR and DRealSR? Since these datasets, to my understanding, also provide multi-scale image data, it would be helpful to clarify whether the intermediate references are taken directly from the dataset or obtained by downsampling the HR target, and if so, with which interpolation method. This detail is important for assessing the reliability of the multi-scale evaluation.

---

> ### Author Rebuttal · Authors · 2026-03-30
>
> > 1. Why reliable decoding of intermediate scales...
>
> Our motivation for enforcing consistency across scales is rooted in the following:
>
> **1. Practical utility:** While prior work addresses multi-scale ISR, doing so efficiently in AR models is unexplored. VAR-based models fail to produce results at intermediate scales, and cannot be deployed for settings where the target zoom is a variable.
>
> **2. Strong inductive bias for efficiency:** The HIT acts as an inductive bias during training. By forcing the model to produce semantically meaningful intermediate representations across scales, our method produces SOTA results with much less parameters than VARSR (310M vs 1B). We attain this relying solely on standard SR datasets, without external data or reward models. Our training is more efficient than VARSR. Please see Table 4. VARSR-d16 w/o our HIT contributions is significantly worse across all datasets and metrics (please see second to last column in Table 4).
>
> **Regarding the visual quality in Fig. 1:** this figure illustrates the capability of our model to work at intermediate scales, something a VAR trained without a hierarchical tokenization can't do (Fig. 1, left). We kindly refer the reviewer to the quantitative results in Tab 4, as well as the results in Fig. 5 and 6, to validate the robustness of our method, which matches or exceeds prior methods in recovering sharp, high-frequency details.
>
> > 2.What evidence that semantic consistency across scales is improved...
>
> Table 2 and Fig. 3 show results for intermediate semantic consistency. Fig. 3 shows that the standard RQVAE doesn’t map intermediate scales to valid images. This is a consequence of the fixed-scale RQ formulation (Sec 3.1).
> **Our formulation is intended to address this limitation.**
> Table 2 shows this quantitatively with results at 128 and 256. On RealSR, VARSR fails at 128 resolution (LPIPs 0.618), and so does the baseline (a VAR w/o HIT, LPIPs 0.686). Those values of LPIPs indicate failure. We report LPIPs of 0.199 at 128 resolution. If VARSR or baseline were producing semantic results, the LPIPs would be much lower. Table 2 supports the evidence of semantic consistency. The qualitative results in Fig. 3 are a visually clear proof of concept.
> As noted, this helps the optimization of the VAR. Without it, tokens are abstract residuals needed to construct the final scale. The network must learn a complex, non-semantic trajectory through the latent space. Enforcing each scale to be decodable drives the network to learn true image-to-image mappings. This is the aforementioned inductive bias: the model needs to first reconstruct low-frequency semantics, add mid-frequency details, and finally generate high-frequency textures. This eases the optimization process, resulting in an efficient VAR producing SOTA results.
>
> > 3. How much of the reported gain …
>
> Our individual contributions are strictly isolated in our current experiments. We do a top to bottom ablation study **by removing each time strictly a single component** from the proposed pipeline, effectively isolating the individual contributions. Table 1 isolates the impact of the RQVAE finetuning; Table 2 isolates the impact of a hierarchical tokenization to train VAR, and Table 3 isolates the impact of using the DPO regularization term. Those results are summarized in the bellow table
>
> | Dataset | HIT | DPO | 128 | 128 | 256 | 256 | 512 | 512 |
> |------|------|------|--------|--------|--------|--------|--------|--------|
> |      |      |      | PSNR   | LPIPS   | PSNR   | LPIPS   | PSNR   | LPIPS   |
> | RealSR   | ✅   | ✅   |   22.09   |   0.1996    |    24.41   |    0.2357   |    25.55   |  0.256|
> | RealSR   | ❌   |✅    |   17.08    |    0.686   |   18.62    |   0.491   |   26.11  |    0.311   |
> | RealSR   | ✅   | ❌   |   20.56   |  0.347    |     23.09  |  0.309   |   25.72  |   0.310    |
>
> > 4. Does the DPO regularization ...
>
> We thank the reviewer for raising this valid concern, which we address through the following two points:
>
> **1. As shown in Table 3,** the DPO consistently improves the PSNR across all scales. PSNR is a strict measure of fidelity, that penalizes hallucinated details. If DPO was acting as a “sharpener” introducing fake details, the PSNR scores would be worse. The improvement in LPIPs and PSNR indicate that DPO is helping the model reconstruct true HR content.
>
> **2. As noted in the paper,** the choice of the LR tokens as the negative sample is intentional. AR-based models applied to image restoration tend to apply a ``shortcut" by simply predicting the conditioning tokens, mainly due to the structural overlap between the input and target images. Our DPO term simply penalizes the "lazy" identity map by pushing the distribution away from the LR tokens. We enforce the model to focus on the actual task of super-resolution rather than simply focus on deblurring the LR images.
>
> > 5. Reference images
>
> The 128 and 256 scales are downsampled from the 512 image.

---

> > ### Author Rebuttal · Reviewer_iQLa · 2026-04-04
> >
> > The inductive bias explanation, together with the quantitative results, makes the motivation more convincing to me and largely addresses my main concern about the link between intermediate-scale decodability and final output quality. My other concerns have also been reasonably addressed by the authors’ clarifications and additional experiments, so I am raising my score to 4.

---

### Decision · Program_Chairs · 2026-04-30

**Decision:**

Accept (regular)

**Comment:**

In the preliminary reviews, two reviewers gave a "Weak Reject" as they had many questions regarding the paper. Although the other two reviewers gave preliminary "Weak Accept" and "Accept" scores, they also raised numerous questions that overlapped with those raised by the first two reviewers.

In addition to several requests for clarifications on specific sentences, the major questions regarding the experiments included:

1. What are the proposed method’s inference latency, FLOPs, FID, and LPIPS compared to existing SOTA methods?

2. What improvements do the HIT and DPO components (ablation study) bring to the intermediate scales?

3. What are the explicit definitions of the terms "Hierarchical VAR" and "Hierarchical RQVAE"?

4. What are the partition sizes, and how do different partition size settings affect performance and inference time?

The authors provided additional numerical evaluation results and explanations addressing the reviewers’ concerns. All four reviewers acknowledged that their questions were fully resolved. Two reviewers raised their scores to "Weak Accept"; ultimately, all four recommended "Weak Accept" or "Accept" scores.

The authors have committed to revising the manuscript to address the reviewers’ concerns and enhance readability. Given the novelty of the proposed method and the promising experimental results, it is anticipated that the authors will successfully improve the manuscript's clarity and reproducibility. Therefore, the paper is recommended for acceptance.